

# RG flows on two-dimensional spherical defects

Tom Shachar$^\star$, Ritam Sinha$^\dagger$ and Michael Smolkin$^\ddagger$

The Racah Institute of Physics, The Hebrew University of Jerusalem, Jerusalem 91904, Israel

$\star$ tom.shachar@mail.huji.ac.il , $\dagger$ ritam.sinha@mail.huji.ac.il , $\ddagger$ michael.smolkin@mail.huji.ac.il

## Abstract

We study two-dimensional spherical defects in $d$-dimensional Conformal Field Theories. We argue that the Renormalization Group (RG) flows on such defects admit the existence of a decreasing entropy function. At the fixed points of the flow, the entropy function equals the anomaly coefficient which multiplies the Euler density in the defect's Weyl anomaly. Our construction demonstrates an alternative derivation of the irreversibility of RG flows on two-dimensional defects. Moreover, in the case of perturbative RG flows induced by weakly relevant deformations, the entropy function decreases monotonically and plays the role of a $C$-function. We provide a simple example to explicitly work out the RG flow details in the proposed construction.



# 1 Introduction

The renormalization group (RG) flow provide a theoretical framework for isolating the degrees of freedom which describe the low-energy phenomena. The idea is to simplify the theory by ignoring its microscopic structure without affecting the low energy physics. In doing so, the number of degrees of freedom decreases, and there has been a long-standing debate about how to quantify this decrease. In 80's, Zamolodchikov formulated and proved the c-theorem, which makes this quantification precise for a wide class of 2-dimensional quantum field theories [1]. Starting from this work many results were obtained in various dimensions [2–20].

In this paper we study RG flows on two-dimensional defects. The defects have a long story, both in two and higher dimensions – see for instance [21–38]. Defect RG flows were also extensively studied in the literature [39–50]. There are a number of exactly established results about the RG flows on line defects [51–54] and their higher dimensional generalizations [55,56]. Recent examples and perturbative calculations in the context of defects cover a wide range of systems and models [57–62]. Here we restrict our attention to the case where the bulk QFT is a $d$-dimensional Euclidean conformal field theory, and the state is simply the flat space vacuum state. We are interested to study RG flows when a two-dimensional spherical defect is present in such a theory. In this setup, the defect changes along the RG flow, but the bulk remains intact. The flow in this case is called a defect RG (DRG) flow.

The full conformal group $SO(d+1,1)$ is broken at the fixed points of DRG. Since a spherical defect is conformally equivalent to a planar one, it preserves the subgroup $SO(3,1)\times SO(d-2)$ of the full conformal group. This symmetry pattern represents global conformal transformations on the two-dimensional planar defect and rotations around it. The theory at the fixed point is called a defect CFT (DCFT).

In what follows, we introduce a renormalized defect entropy which is fixed by the characteristic size of the defect. Our construction is similar to the one previously employed in the context of entanglement entropy [63]. For a DCFT, it reduces to the dimensionless "central charge" that multiplies the Euler density in the defect's Weyl anomaly, whereas for a general quantum field theory, it interpolates between the central charges of the UV and IR fixed points as the radius of the spherical defect is varied from zero to infinity. Using the ideas introduced in [53], we show that the renormalized defect entropy necessarily decreases from its initial value along the DRG flow, thus providing an alternative proof for irreversibility of the DRG flows on two-dimensional defects [55]. Furthermore, we argue that when the DRG flow is induced by a sufficiently weak relevant deformation of the UV fixed point, the renormalized defect entropy exhibits monotonic decrease and plays the role of a $C$-function throughout all orders in perturbation theory.

The paper is organized as follows. In section 2 we review the derivation of the Ward identities which are necessary for our needs. In section 3 we define the renormalized entropy function, and use it to reproduce the sum rule as well as prove the irreversibility of DRG flows on two-dimensional defects. In section 4 we provide an instructive example which explicitly illustrates various details of DRG flows discussed in this paper. We conclude in section 5.

# 2 Ward identities

As a starting point, we review a higher dimensional generalization of the identities obtained in [53]. Consider a $p$-dimensional defect $\mathcal{D}$, embedded in a $d$-dimensional Euclidean bulk. For simplicity, the bulk is assumed to be flat. The theory is governed by a DCFT action perturbed

by a set of relevant defect operators $\mathcal{O}_i$ with scaling dimensions $\Delta_i < p$,

$$I = I_{\text{DCFT}} + g^i \int_{\mathcal{D}} d^p \sigma \sqrt{\hat{\gamma}} \, \mathcal{O}_i \,, \tag{1}$$

where $\hat{\gamma}_{ac}$ is the induced metric on the defect.

The bulk and defect stress-tensors, $T^{\mu\nu}$, $\hat{T}^{\mu\nu}$ and the displacement operator $D_\mu$ are defined through the variation of the effective action, $W$, with respect to the bulk metric $g_{\mu\nu}$[1] and embedding function $X^\mu(\sigma)$,

$$\delta W = -\frac{1}{2} \int_{\mathcal{M}} d^d x \sqrt{g} \, \delta g_{\mu\nu} \langle T^{\mu\nu} \rangle + \int_{\mathcal{D}} d^p \sigma \sqrt{\hat{\gamma}} \, \delta X^\mu(\sigma) \langle D_\mu \rangle$$
$$-\frac{1}{2} \int_{\mathcal{D}} d^p \sigma \sqrt{\hat{\gamma}} \left[ \delta g_{\mu\nu} \langle \hat{T}^{\mu\nu} \rangle + \dots \right] . \tag{2}$$

The total stress tensor, $T^{\text{tot}}_{\mu\nu}$ is defined by,

$$T^{\text{tot}}_{\mu\nu} = T_{\mu\nu} + \hat{T}_{\mu\nu} \delta_{\mathcal{D}} \,, \tag{3}$$

where $\delta_{\mathcal{D}}$ denotes the delta function which restricts the bulk integrals to the defect, *i.e.*, by definition $\int d^d x \delta_{\mathcal{D}} = \int_{\mathcal{D}} d^2 \sigma \sqrt{\hat{\gamma}}$, or equivalently, the integral of the $d$-dimensional delta-function over the defect satisfies, $\int_{\mathcal{D}} \delta = \delta_{\mathcal{D}}$. By assumption, the bulk theory is conformal, therefore $T^\mu_\mu = 0$.

In what follows, the indices $a, b, ..$ will be used to denote the $p$ tangential directions $e^\mu_a = \frac{\partial X^\mu}{\partial \sigma^a}$. Similarly, the indices $I, J, ..$ will be used to denote the $d - p$ normal vectors, $n^\mu_I$.

The three physical quantities in (2) are related by Ward identities associated with the invariance of $W$ under the bulk and defect reparametrizations. In the former case, the condition $\delta W = 0$ is imposed under an infinitesimal diffeomorphism of the form $x^\mu \to x'^\mu = x^\mu - \xi^\mu$,

$$\delta x^\mu = -\xi^\mu \,, \quad \delta g_{\mu\nu} = \nabla_\mu \xi_\nu + \nabla_\nu \xi_\mu \,. \tag{4}$$

By splitting the bulk into normal and tangential components, $\xi_\nu = e^a_\nu \xi_a + n^I_\nu \xi_I$, this gives the following Ward identity (See Appendix A for details)

$$\nabla_\mu T^{\mu\nu} \xi_\nu + \delta_{\mathcal{D}} \left[ \left( \hat{\nabla}_b \hat{T}^{ba} - D^a \right) \xi_a + \left( \nabla_a \hat{T}^{aI} - D^I - K^I_{ab} \hat{T}^{ab} \right) \xi_I \right] = 0 \,, \tag{5}$$

where $\hat{\nabla}_a$ is the covariant derivative on the defect.[2]

Likewise, the same condition, $\delta W = 0$, is imposed for infinitesimal reparametrizations of the defect,[3]

$$\delta \sigma^a = -\zeta^a \,, \quad \delta X^\mu = e^\mu_a \zeta^a \,, \quad \delta g_{\mu\nu} = 0 \,. \tag{6}$$

---

[1] We adhere to the conventions of [55, 64], see [29] for alternative definitions. The ellipsis in the last line of (2) encode variations associated with the normal derivatives of the bulk metric. These terms are irrelevant for the low dimensional defects considered in this paper. Moreover, in general the first-order normal derivative terms have no impact on the results for spherical defects.

[2] The symbol $\hat{\nabla}$ is used to denote the covariant derivative on the defect, compatible with the induced metric, $\hat{\nabla}_a \hat{\gamma}_{cb} = 0$. Refer also to the paragraph below (A.4).

[3] Finite reparametrizations of the defect are defined as follows,

$$\tilde{\sigma}^a = \tilde{\sigma}^a(\sigma), \, \tilde{X}^\mu(\tilde{\sigma}) = X^\mu(\sigma), \quad \tilde{g}_{\mu\nu}(\tilde{X}(\tilde{\sigma})) = g_{\mu\nu}(X(\sigma)) = g_{\mu\nu}(\tilde{X}(\tilde{\sigma})).$$

Employing this definition yields the conventional transformation for the induced metric on the defect,

$$\tilde{h}_{ac}(\tilde{\sigma}) = \frac{\partial \tilde{X}^\mu(\tilde{\sigma})}{\partial \tilde{\sigma}^a} \frac{\partial \tilde{X}^\nu(\tilde{\sigma})}{\partial \tilde{\sigma}^c} \tilde{g}_{\mu\nu}(\tilde{X}(\tilde{\sigma})) = \frac{\partial \sigma^b}{\partial \tilde{\sigma}^a} \frac{\partial \sigma^d}{\partial \tilde{\sigma}^c} h_{bd}(\sigma).$$

The infinitesimal diffeomorphism, $\tilde{\sigma}^a = \sigma^a - \zeta^a$, takes the form given in (6).

Combined with (2), they imply $D_a \equiv e_a^\mu D_\mu = 0$, *i.e.,* tangential displacements are trivial.
Now consider a dimensionless dilaton background, $\Phi(\sigma)$, localized at the defect [53]. By definition, the dilaton couples linearly to the trace of the defect stress-tensor, *i.e.,* $\hat{T} \equiv g^{\mu\nu}\hat{T}_{\mu\nu} = \frac{1}{\sqrt{\hat{\gamma}}}\frac{\delta W}{\delta \Phi}$, and under defect reparameterizations, it transforms as,

$$\delta\Phi(\sigma) = -\zeta^a \partial_a \Phi(\sigma) . \tag{7}$$

Hence, (2) takes the form

$$0 = \delta W = \int_{\mathcal{D}} d^p\sigma \sqrt{\hat{\gamma}} \, \delta X^\mu(\sigma) D_\mu + \int_{\mathcal{D}} d^p\sigma \frac{\delta W}{\delta\Phi}\delta\Phi = \int_{\mathcal{D}} d^p\sigma \sqrt{\hat{\gamma}} \, \zeta^a \left(D_a - \hat{T}\partial_a\Phi\right) .$$

As a result, in the presence of dilaton, we have $D_a = \hat{T}\partial_a\Phi$.

A bulk CFT in $d$-dimensions is invariant under the full conformal group $SO(d+1,1)$. However, the conformal defect partially breaks it. In particular, a $p$-dimensional *spherical conformal defect* is invariant under the subgroup $SO(p+1,1) \times SO(d-p) \subset SO(d+1,1)$. The first factor represents conformal group of the $p$-sphere. For $p=2$, it has six generators, $SO(3,1) \simeq SL(2,\mathbb{C})$. This group acts on the 3-dimensional ambient subspace hosting the sphere. For simplicity, we parametrize this subspace by $x^\mu$, $\mu = 1,2,3$. The group $SO(d-2)$, generates transformations in the transverse directions to the defect. In the conformal frame where the defect is planar, $SO(d-2)$ represents ordinary rotations in the transverse space to a flat defect. The six $SL(2,\mathbb{C})$ conformal Killing vectors are given by,

$$\xi^\mu_{(a)} = \frac{1}{2}\left[\delta^\mu_a\left(R + \frac{x^2}{R}\right) - 2\frac{x_a x^\mu}{R}\right], \qquad \chi^\mu_{(a)} = \delta^\mu_b \epsilon_a^{\ bc} x_c, \qquad a,b,c = 1,2,3, \tag{8}$$

where $R$ is the radius of the sphere. The $\xi$'s are particular combinations of translations and special conformal transformations which preserve the sphere, whereas $\chi$'s represent rotations. It can be checked that (8) satisfy the conformal Killing equations in the bulk as well as on the defect. Each such Killing vector gives rise to a conserved charge in the bulk,

$$Q_\xi = \int_\Sigma d^{d-1}x \, \Sigma_\mu T^{\mu\nu}\xi_\nu, \tag{9}$$

where $\Sigma$ is a hypersurface.

Now consider the vacuum expectation value of $Q_\xi$ provided that the hypersurface $\Sigma$ wraps around the spherical defect. By definition, when $Q_\xi$ surrounds an operator, it transforms it, *i.e.,*

$$\langle Q_\xi \rangle = \langle \delta_\xi \mathcal{D} \rangle, \tag{10}$$

where $\delta_\xi\mathcal{D}$ is a small change in the spherical defect induced by the conformal Killing vectors (8). This change vanishes if the defect is conformal (DCFT), but we do not assume it in what follows. In fact, the scale invariance is broken in the presence of fixed dilaton background.

The boundary conditions at infinity correspond to a conformal vacuum state. Since $Q_\xi$ annihilates it, and there are no other insertions in the path integral save the defect, we deduce that for any $\Phi(\sigma)$,

$$0 = \langle \delta_\xi \mathcal{D} \rangle = \langle Q_\xi \rangle = \int_\Sigma d^{d-1}x \, \Sigma_\mu \langle T^{\mu\nu} \rangle \xi_\nu = -\int_{\mathcal{D}} d^2\sigma \left\langle \hat{\nabla}_b \hat{T}^{ba} - \hat{T}\partial^a\Phi \right\rangle \xi_a, \tag{11}$$

where in the last equality we used Gauss's theorem followed by (5) with $\xi^I = 0$ for the Killing vectors (8), as well as tracelessness of the bulk stress tensor. Integrating by parts, yields

$$0 = \langle \delta_\xi \mathcal{D} \rangle = \int d^2\sigma \left\langle \hat{T}^{ba}\hat{\nabla}_b\xi_a + \hat{T}\partial^a\Phi \, \xi_a \right\rangle. \tag{12}$$

Next, recall that the conformal Killing vectors satisfy

$$\hat{\nabla}_a \xi_b + \hat{\nabla}_b \xi_a = \frac{2}{p}\left(\hat{\nabla}\cdot\xi\right)\hat{\gamma}_{ab}\bigg|_{p=2} = \left(\hat{\nabla}\cdot\xi\right)\hat{\gamma}_{ab}. \tag{13}$$

Combining, we finally obtain

$$0 = \langle\delta_\xi \mathcal{D}\rangle = \int d^2\sigma\left(\frac{1}{2}\left(\hat{\nabla}\cdot\xi\right) + \partial^b \Phi \xi_b\right)\langle\hat{T}\rangle. \tag{14}$$

The dilaton couples linearly to the trace of the defect stress tensor, and therefore the right hand side of (14) can be interpreted as a change in the defect due to a small variation in the dilaton profile, $\delta\Phi \sim \frac{1}{2}\left(\hat{\nabla}\cdot\xi\right) + \partial^b \Phi \xi_b$. In particular, it follows from (14) that one can identify two defects if their dilaton backgrounds are related by[4]

$$\Phi \sim \Phi + \alpha\left(\frac{1}{p}\hat{\nabla}_a \xi^a + \xi^a \partial_a \Phi\right), \quad \alpha \ll 1. \tag{15}$$

Hence,

$$\log Z_\Phi = \log Z_{\Phi+\delta\Phi} = \log Z_\Phi + \int_{\mathcal{D}} d^2\sigma\langle\hat{T}(\sigma)\rangle_\Phi \delta\Phi(\sigma) + \mathcal{O}\left(\delta\Phi^2\right), \tag{16}$$

with $\delta\Phi$ defined in (15). Expanding around $\Phi = 0$ results in a series of constraints. At $\mathcal{O}(\Phi^0)$ we have,

$$\int d^p\sigma\,\frac{1}{p}\left(\hat{\nabla}_a \xi^a\right)\langle\hat{T}(\sigma)\rangle_0 = 0, \tag{17}$$

and at $\mathcal{O}(\Phi^1)$,

$$\int d^p\sigma\,\xi^a \partial_a \Phi\langle\hat{T}(\sigma)\rangle_0 + \int d^p\sigma_1 d^p\sigma_2\,\frac{1}{p}\left(\hat{\nabla}\cdot\xi(\sigma_1)\right)\Phi(\sigma_2)\langle\hat{T}(\sigma_1)\hat{T}(\sigma_2)\rangle_0 = 0. \tag{18}$$

For our purposes (18) is enough, and we ignore all the other identities.

Notice that the covariant derivative takes a simple form for the first three Killing vectors in (8), i.e., $\hat{\nabla}\xi^a(\theta,\phi) = -2n^a(\theta,\phi)$. where $\hat{n} = (\sin\theta\cos\phi, \sin\theta\sin\phi, \cos\theta)$ is a unit vector. Choosing a dilaton profile of the form, $\Phi(\theta,\phi) \equiv n^b(\theta,\phi)$ for any $b = 1, 2, 3$, and introducing the following notation $\int d^2\sigma\sqrt{\hat{\gamma}} \equiv \int_{S^2}$ for brevity, the double integral in (18) becomes,

$$I^{ab} = \int_{S^2}\int_{S^2} n_1^a\, n_2^b\langle\hat{T}(\hat{n}_1)\,\hat{T}(\hat{n}_2)\rangle. \tag{19}$$

Due to the $SO(3)$ invariance of the integration measure and the two-point function, we deduce that $I^{ab}$ is an invariant bulk tensor, and therefore it is proportional to $\delta^{ab}$,

$$I^{ab} = \frac{1}{3}\delta^{ab}\int_{S^2}\int_{S^2}(\hat{n}_1\cdot\hat{n}_2)\langle\hat{T}(\hat{n}_1)\,\hat{T}(\hat{n}_2)\rangle. \tag{20}$$

Setting $a = b = 3$, i.e., $\Phi \propto \cos\theta$, and evaluating the first term in (18), yields

$$\int_{S^2}\left\langle\hat{T}(\hat{n})\right\rangle = \frac{1}{2}\int_{S^2}\int_{S^2}(\hat{n}_1\cdot\hat{n}_2)\langle\hat{T}(\hat{n}_1)\,\hat{T}(\hat{n}_2)\rangle. \tag{21}$$

In fact, a similar expression also holds for higher dimensional spherical defects.

---

[4]Our analysis also holds in the case of a $p$-dimensional spherical defect. This is why a general $p$ appears in (15).

# 3 Irreversibility of the DRG flows

In this section we establish the irreversibility of DRG flows on the two-dimensional defects through the use of renormalized defect entropy defined below. It is derived from the defect $\mathcal{F}$-function defined by

$$\mathcal{F} = -\log \frac{Z_{\mathcal{D}}}{Z_{\text{CFT}}}, \tag{22}$$

where $Z_{\text{CFT}}$ is the partition function of an ambient CFT without defect. The $\mathcal{F}$-function is dimensionless, and therefore it depends on the dimensionless couplings and dimensionless combination $\mu R$, where $\mu$ is the floating cutoff scale.[5]

For a 2$d$ defect of characteristic size $R$ embedded in a flat Euclidean space, the $\mathcal{F}$-function at the UV fixed point of the RG flow takes the form[6]

$$\mathcal{F}^{\text{UV}}_{\text{DCFT}} = c_0 + \frac{a_0 \mu_{\text{UV}}^2}{4\pi} \int d^2\sigma \sqrt{\hat{\gamma}} \tag{23}$$
$$- \left( \frac{b_0}{24\pi} \int d^2\sigma \sqrt{\hat{\gamma}} \, \mathcal{R} + \frac{b_1}{24\pi} \int d^2\sigma \sqrt{\hat{\gamma}} \, \text{Tr}\big( \tilde{K}_\mu \tilde{K}^\mu \big) \right) \log(\mu_{\text{UV}} R).$$

Here, $\mathcal{R}$ is the Ricci scalar of the defect, whereas $\tilde{K}^\mu_{ac} = K^\mu_{ac} - \frac{1}{2}\hat{\gamma}_{ac}\text{Tr}(K^\mu)$ is the traceless part of the defect extrinsic curvature $K^\mu_{ac}$. The constants in the above expression are functions of the critical couplings. This ansatz is obtained by solving the Wess-Zumino consistency conditions at the fixed points of the DRG flow [66–68]. Moreover, for a sphere in flat space $\tilde{K}^\mu_{ac} = 0$, and therefore (23) simplifies

$$\mathcal{F}^{\text{UV}}_{\text{DCFT}}\Big|_{S^2} = c_{\text{UV}} + a_{\text{UV}}(\mu_{\text{UV}} R)^2 - \frac{b_{\text{UV}}}{3} \log(\mu_{\text{UV}} R), \tag{24}$$

where $b_{\text{UV}} = b_0$.

The $\mathcal{F}$-function changes if a UV DCFT is subject to a relevant deformation. However, the precise structure of the $\mathcal{F}$-function away from the UV fixed point is not essential for our needs. Our analysis relies on the existence of a cutoff scale $\mu_{\text{IR}} \ll \mu_{\text{UV}}$, where the theory is controlled by the IR DCFT, and the $\mathcal{F}$-function can be recast as (24) with $\mu_{\text{UV}}, c_{\text{UV}}, a_{\text{UV}}$ and $c_{\text{UV}}$ replaced by their IR counterparts.

The first two terms in (24) are non-universal, because one can shift $c_{\text{UV}}$ by rescaling $\mu_{\text{UV}}$, whereas $a_{\text{UV}}$ can be arbitrarily changed by adding a finite local counterterm to the defect (cosmological constant). In contrast, $b_{\text{UV}}$ does not suffer from the ambiguities, it is universal and satisfies $b_{\text{UV}} \geq b_{\text{IR}}$ for the UV and IR ends of the RG trajectory [55].[7] In what follows, we provide an alternative derivation of this inequality. Moreover, we prove that our construction decreases monotonically along the DRG flows induced by weakly relevant deformations of the UV DCFT leading to a $C$-function. To the best of our knowledge, this is the first perturbative example of a $C$-function in the context of two-dimensional defects.[8]

To isolate the universal part of $\mathcal{F}$, we define *the renormalized defect entropy* (RDE) as follows

$$S = -R\partial_R \left( 1 - \frac{1}{2}R\partial_R \right) \mathcal{F} = \frac{1}{2} \left( R^2 \partial_R^2 - R\partial_R \right) \mathcal{F}. \tag{25}$$

This definition is motivated by the so-called Renormalized Entanglement Entropy in four spacetime dimensions [63]. The derivatives with respect to $R$ are designed to eliminate the non-universal terms, such that $S = \frac{b_0}{3}$ at the fixed points of the DRG flows on spherical defects.

---

[5]By introducing suitable explicit factors of $\mu$ all couplings may be assumed to be dimensionless.

[6]There are additional contributions if the ambient Euclidean space is curved, see *e.g.,* [65, 66].

[7]$b_1$ is also universal, but we do not study it in this work.

[8]See also [69], where the ideas of entanglement [9, 45, 52] are used to build a proposal for the $C$-function.

The Renormalized Entanglement Entropy is neither monotonic nor proved to be useful in establishing irreversibility of RG flows in four dimensions [63]. In contrast, as shown below, the RDE introduced in (25) necessarily decreases between the two ends of the DRG flow.

Introducing a constant dilaton profile, one can rewrite (25) in an equivalent form

$$S = \left[ \left( \frac{1}{2} \frac{d^2}{d\Phi^2} - \frac{d}{d\Phi} \right) \mathcal{F} \right]_{\Phi=0} = \int_{S^2} \langle \hat{T}(\sigma) \rangle - \frac{1}{2} \int_{S^2} \int_{S^2} \langle \hat{T}(\sigma_1) \hat{T}(\sigma_2) \rangle \,, \tag{26}$$

because by definition the dilaton is coupled to the trace of the defect stress tensor. Using the constraint equation (21), we get

$$S = -\frac{1}{4R^2} \int_{S^2} \int_{S^2} s^2(\sigma_1, \sigma_2) \langle \hat{T}(\sigma_1) \hat{T}(\sigma_2) \rangle \,, \tag{27}$$

where $s^2(\sigma_1, \sigma_2) = 2R^2(1 - \hat{n}_1 \cdot \hat{n}_2)$ is the square of the chordal distance between the two points $\sigma_1$ and $\sigma_2$ on the surface of a two-dimensional sphere.

Note that (27) necessarily includes the contribution of the contact term, otherwise (21) is not satisfied at the fixed points of the DRG flow, where the trace of defect stress tensor vanishes up to an anomaly. In particular, while the two-point function is positive definite due to unitarity of the theory, the contact term does not have a definite sign. Hence, the RDE is not necessarily positive.

To isolate the contribution of the contact term, we evaluate (27) at the UV fixed point of the DRG flow. To this end, we note that the UV DCFT satisfies,

$$\langle \hat{T} \rangle_{\text{UV}} = \frac{b_{\text{UV}}}{24\pi} \mathcal{R} \quad \Rightarrow \quad \langle \hat{T}(\sigma_1) \hat{T}(\sigma_2) \rangle_{\text{UV}} = -\frac{b_{\text{UV}}}{12\pi} (\mathcal{R} + \nabla^2) \frac{\delta(\sigma_1, \sigma_2)}{\sqrt{\hat{\gamma}(\sigma_1)}} \,, \tag{28}$$

where the contact term on the right is obtained by varying the anomaly term on the left with respect to the induced metric on the defect. Substituting this expression into (27), yields the expected result $S_{\text{UV}} = \frac{b_{\text{UV}}}{3}$. In particular, (27) can be written as follows

$$S = \frac{b_{\text{UV}}}{3} - \frac{1}{4R^2} \int_{S^2} \int_{S^2} s^2(\sigma_1, \sigma_2) \langle \hat{T}(\sigma_1) \hat{T}(\sigma_2) \rangle = \frac{b_{\text{UV}}}{3} - \pi \int_{S^2} s^2(\sigma) \langle \hat{T}(\sigma) \hat{T}(0) \rangle \,, \tag{29}$$

where the contact term is now excluded from the positive definite $\langle \hat{T} \hat{T} \rangle$. In the second equality we used invariance of the integrand under rotations of $S^2$ to position $\sigma_2$ at the south pole of the sphere ($\sigma_2 = 0$).

The integral on the right hand side of (29) is manifestly positive and finite, because the sphere introduces a natural IR cut off, whereas the limit of coincident points, $\sigma \to 0$, is dominated by the UV DCFT with vanishing $\hat{T}$, i.e., $\langle \hat{T} \hat{T} \rangle = \beta^i \beta^j \langle \mathcal{O}_i \mathcal{O}_j \rangle$, where $\beta^i$'s are the beta functions of various couplings, whereas the *renormalized* operators $\mathcal{O}_i$ are associated with the relevant deformations of the UV fixed point. Hence, $\langle \hat{T} \hat{T} \rangle$ is less singular than $1/s^4$, and the integral converges in this limit.

The RDE is a function of dimensionless couplings, $g^i$, and $\mu R$. The natural choice for the running scale is $\mu \sim 1/R$,

$$S\big(\mu R, \ g^i(\mu)\big)\Big|_{\mu \sim 1/R} = S\big(g^i(R^{-1})\big) \,. \tag{30}$$

Thus the value of $S$ along the RG trajectory can be probed by varying the radius of the sphere. In particular, taking the limit $R \to \infty$, we establish the irreversibility of DRG flows on the two-dimensional defects

$$\frac{b_{\text{IR}} - b_{\text{UV}}}{3} = -\pi \int_{S^2} s^2(\sigma) \langle \hat{T}(\sigma) \hat{T}(0) \rangle \Big|_{R \to \infty} \leq 0 \quad \Leftrightarrow \quad b_{\text{IR}} \leq b_{\text{UV}} \,. \tag{31}$$

The RDE might not be necessarily monotonic along the RG trajectory. To show it explicitly, let us differentiate (30) with respect to $R$ and use $\hat{T} = \beta^i \mathcal{O}_i$,[9]

$$R\frac{d}{dR}S(g^i) = -\beta^i \frac{\partial}{\partial g^i}S(g^i) = +\pi\beta^i\frac{\partial}{\partial g^i}\int_{S^2} s^2(\sigma)\beta^j\beta^k\langle\mathcal{O}_j(\sigma)\mathcal{O}_k(0)\rangle$$

$$= \pi\beta^i\beta^j\Big(2\frac{\partial\beta^k}{\partial g^i} + \beta^k\frac{\partial}{\partial g^i}\Big)\int_{S^2} s^2(\sigma)\langle\mathcal{O}_j(\sigma)\mathcal{O}_k(0)\rangle = -2\pi^2\beta^i\beta^j h_{ij}. \tag{34}$$

In the last equality we have defined the matrix $h_{ij}(g)$, which is analogous to the well-known Zamolodchikov metric [1].

The beta functions vanish at the fixed points of the DRG flow, therefore $\partial\beta^j/\partial g^i$ likewise the first term within parenthesis in (34) necessarily flip the sign along the flow. Similarly, the second term within parenthesis does not exhibit a definite sign, because it explicitly depends on the three point function. The upshot of this discussion is that the positive definiteness of $h_{ij}$ and, consequently, the monotonicity of $S$ is not evident. That said, the renormalized defect entropy is monotonic for a large class of RG flows, as demonstrated in the next subsection.

## 3.1 Perturbative DRG flow

Consider a DCFT perturbed by a set of weakly relevant defect operators $\mathcal{O}_i$ with scaling dimensions $\Delta_i = 2 - \epsilon_i$ where $0 < \epsilon_i \ll 1$. We choose the operators $\mathcal{O}_i$ such that at the UV fixed point they satisfy

$$\langle\mathcal{O}_i(\sigma_1)\mathcal{O}_j(\sigma_2)\rangle_{\text{UV}} = \frac{\delta_{ij}}{s(\sigma_1,\sigma_2)^{2\Delta_i}}, \tag{35}$$

$$\langle\mathcal{O}_i(\sigma_1)\mathcal{O}_j(\sigma_2)\mathcal{O}_k(\sigma_3)\rangle_{\text{UV}} = \frac{C_{ijk}}{s(\sigma_1,\sigma_2)^{\Delta_i+\Delta_j-\Delta_k}s(\sigma_1,\sigma_3)^{\Delta_i+\Delta_k-\Delta_j}s(\sigma_2,\sigma_3)^{\Delta_j+\Delta_k-\Delta_i}}.$$

The above weakly relevant deformations give rise to a perturbative RG flow of the form [2, 70],[10]

$$\beta^i = \mu\frac{dg^i}{d\mu} = -\epsilon_i g^i + \pi C^i_{jk}g^j g^k + \mathcal{O}(g^3), \tag{36}$$

where the indices are raised and lowered with the Kronecker delta. In particular, the IR fixed point is located in the vicinity of the UV DCFT and one can use conformal perturbation theory to calculate $\Delta b = b_{\text{IR}} - b_{\text{UV}}$. Substituting $\hat{T} = \beta^i \mathcal{O}_i$ into (31) and expanding around the UV fixed point, yields

$$\Delta b = -3\pi\beta^i\beta^j\int d^2\sigma\sqrt{\hat{\gamma}}\,s^2(\sigma)\Big(Z_i{}^k Z_j{}^\ell\langle\mathcal{O}_k(\sigma)\mathcal{O}_\ell(0)\rangle_{\text{UV}}$$

$$-g^k\int d^2\sigma'\sqrt{\hat{\gamma}}\langle\mathcal{O}_i(\sigma)\mathcal{O}_j(0)\mathcal{O}_k(\sigma')\rangle_{\text{UV}} + \mathcal{O}(g^2)\Big), \tag{37}$$

---

[9]We drop the anomaly term from $\hat{T}$, because it does not contribute to the connected correlator in (29). Note also that the flow equation for $S(\mu R,\,g^i(\mu))$ can be derived from the Callan-Symanzik equation,

$$\Big(\mu\frac{\partial}{\partial\mu} + \beta^i\frac{\partial}{\partial g^i}\Big)S(\mu R,\,g^i(\mu)) = 0. \tag{32}$$

The differential operator within parenthesis in (32) commutes with $R\frac{\partial}{\partial R}$, and therefore (32) follows from the definition of $S$ and the Callan-Symanzik equation for the $\mathcal{F}$-function. Hence,

$$-\beta^i\frac{\partial}{\partial g^i}S(g^i) = \mu\frac{\partial}{\partial\mu}\Big|_{\mu\sim 1/R}S(\mu R,\,g^i(\mu)). \tag{33}$$

[10]See also next section.

where $Z_i{}^k$ is the mixing matrix, which relates the renormalized $\mathcal{O}_i$ to its UV counterpart, $\mathcal{O}_i = Z_i{}^k \mathcal{O}_k^{\text{UV}}$. We keep only linear terms within parenthesis in the above expression, because the perturbative beta functions (36) are evaluated up to $\mathcal{O}(g^2)$.

For simplicity consider the case with equal $\epsilon_i$'s, then

$$Z_{ij} = \delta_{ij} + \frac{2\pi C_{ijk} g^k}{\epsilon} + \mathcal{O}(g^2). \tag{38}$$

Using (35), (36), (B.5) and (B.15), we obtain

$$\Delta b = -3\pi^2 \epsilon\, \delta_{ij}\, g_{\text{IR}}^i\, g_{\text{IR}}^j + 2\pi^3 C_{ijk}\, g_{\text{IR}}^i\, g_{\text{IR}}^j\, g_{\text{IR}}^k + \mathcal{O}(g_{\text{IR}}^4) = -\pi^2 \epsilon\, \delta_{ij}\, g_{\text{IR}}^i\, g_{\text{IR}}^j + \mathcal{O}(g_{\text{IR}}^4) < 0, \tag{39}$$

where the couplings $g_{\text{IR}}^i$ correspond to the IR fixed point of the DRG flow, and we used $\beta^i(g_{\text{IR}})=0$ in the second equality to simplify the expression. If there is only one relevant deformation, *i.e.,* a single coupling $g_{\text{IR}}$ and one OPE coefficient $C_1$, we obtain

$$\beta(g_{\text{IR}}) = 0 \quad \Rightarrow \quad g_{\text{IR}} = \frac{\epsilon}{\pi C_1} \quad \Rightarrow \quad \Delta b = -\frac{\epsilon^3}{C_1^2} < 0. \tag{40}$$

Lastly, the matrix $h_{ij}$ in (34) is given by[11]

$$h_{ij} = \delta_{ij} + \mathcal{O}(g). \tag{41}$$

Therefore, as long as the perturbative expansion remains valid, $h_{ij}$ retains its positive definiteness within a small neighborhood of the UV DCFT. Specifically, the RDE exhibits perturbative monotonicity to all orders in the coupling constant and serves as a *C*-function, provided that the DRG flow is generated by a sufficiently weak relevant deformation of the UV DCFT.

## 4 Example of the DRG flow

In this section we present a concrete and simple example of a DCFT, where the general concepts of the previous sections can be explicitly illustrated. With this aim, consider a free massless scalar field in a $d$-dimensional Euclidean bulk coupled to a two-dimensional defect $\mathcal{D}$,

$$I = \frac{1}{2}\int d^d x\, \partial_\mu \phi_0 \partial^\mu \phi_0 + g_0 \int_{\mathcal{D}} d^2\sigma \sqrt{\hat{\gamma}}\, \phi_0^2 + \int_{\mathcal{D}} d^2\sigma \sqrt{\hat{\gamma}}\Big(\Lambda_0 - \frac{b_0}{24\pi}\mathcal{R}\Big), \tag{42}$$

where the last integral on the right hand side represents the geometric counterterms with $\Lambda_0$ and $\mathcal{R}$ being the cosmological constant and Ricci scalar of the induced metric respectively. This action is Gaussian with a space-time dependent mass term of the form, $m^2 = 2g_0 \delta_{\mathcal{D}}$. We employ the minimal subtraction scheme to absorb the divergences due to the presence of a singular mass term.

Varying (42), yields

$$T_{\mu\nu}^{\text{tot}} = \partial_\mu \phi_0 \partial_\nu \phi_0 - \frac{1}{2}\delta_{\mu\nu}(\partial \phi_0)^2 - \frac{d-2}{4(d-1)}(\partial_\mu \partial_\nu - \delta_{\mu\nu}\partial^2)\phi_0^2 - \hat{\gamma}_{\mu\nu}(g_0\phi_0^2 + \Lambda_0)\delta_{\mathcal{D}}, \tag{43}$$

where the third term on the right hand side represents the well known improvement in the bulk, and we used the following identities,

$$\delta\hat{\gamma}^{ac} = \delta g^{\mu\nu} e_\mu^a e_\nu^c, \quad \hat{\gamma}_{\mu\nu} = \hat{\gamma}_{ac} e_\mu^a e_\nu^c.$$

---

[11]The first equality in (39) is applicable to any coupling $g^i$ along the DRG flow. Consequently, one simply applies $-\beta^i \frac{\partial}{\partial g^i}$ to the expression for $\Delta b/3$ to derive $h_{ij}$.

Taking trace of $T_{\mu\nu}^{\text{tot}}$, and using (3) along with the tracelessness of the bulk stress tensor, $T \equiv T_\mu^\mu = 0$, yields

$$T_{\text{tot}} = \hat{T}\,\delta_{\mathcal{D}} = \frac{d-2}{2}\phi_0\partial^2\phi_0 - 2\big(g_0\phi_0^2 + \Lambda_0\big)\delta_{\mathcal{D}} = (d-4)g_0\phi_0^2\delta_{\mathcal{D}} - 2\Lambda_0\delta_{\mathcal{D}}\,, \qquad (44)$$

where the last equality follows from the equation of motion operator,

$$E = -\partial^2\phi_0 + 2g_0\phi_0\,\delta_{\mathcal{D}} = 0\,. \qquad (45)$$

Hence, we finally obtain,

$$\hat{T} = (d-4)g_0\phi_0^2 - 2\Lambda_0\,. \qquad (46)$$

To facilitate further analysis, we assume that $d = 4 - \epsilon$, *i.e.*, the bare coupling $g_0$ is weakly relevant. In particular, conformal perturbation theory can be employed to relate $g_0$ to the renormalized coupling $g$ at an arbitrary energy scale $\mu$. To this end, we note that up to second order in $g_0$ the defect insertion in the path integral can be written as follows,

$$e^{-g_0\int_{\mathcal{D}}\phi_0^2} = 1 - g_0\int_{\mathcal{D}}\phi_0^2(\sigma_1) + \frac{g_0^2}{2}\int_{\mathcal{D}}\int_{\mathcal{D}}\phi_0^2(\sigma_1)\phi_0^2(\sigma_2) + \dots \qquad (47)$$

All scales are included in the above expression. To get the defect at a given scale $\mu$, we integrate out the distances in the range $0 \leq \ell \leq \mu^{-1}$. This calculation boils down to excluding a small ball of radius $\mu^{-1}$ around $\phi_0^2(\sigma_1)$ in the second term on the right hand side of (47),

$$\int_{\mathcal{D}}\phi_0^2(\sigma_1)\phi_0^2(\sigma_2) = \int_{\mathcal{D}}^{\sigma_{12}>\mu^{-1}}d^2\sigma_2\sqrt{\hat{\gamma}}\,\phi_0^2(\sigma_1)\phi_0^2(\sigma_2) + \int_{\mathcal{D}}^{0\leq\sigma_{12}\leq\mu^{-1}}d^2\sigma_2\sqrt{\hat{\gamma}}\,\frac{C\,\phi_0^2(\sigma_1)}{\sigma_{12}^{d-2}} + \dots\,, \qquad (48)$$

where $\sigma_{12}$ is the distance between the points $\sigma_1$ and $\sigma_2$ in the $d$-dimensional Euclidean space, and $C$ is the OPE coefficient at the Gaussian fixed point,

$$\phi_0^2(x)\phi_0^2(0) \sim \frac{C}{|x|^{d-2}}\,\phi_0^2(0) + \dots \qquad (49)$$

The last term in (48) contributes to the renormalization of the coupling $g_0$ in (47),

$$g(\mu)\mu^\epsilon = g_0 - \frac{\pi C}{\epsilon}g_0^2\mu^{-\epsilon} + \mathcal{O}(g_0^3) \quad \Rightarrow \quad g_0 = g\mu^\epsilon\Big(1 + \frac{\pi C g}{\epsilon} + \mathcal{O}(g^2)\Big)\,, \qquad (50)$$

where $g$ is the dimensionless running coupling constant. Thus,

$$\beta = \mu\frac{dg}{d\mu} = -\epsilon g + \pi C g^2 + \mathcal{O}(g^3)\,. \qquad (51)$$

Furthermore, the renormalized defect operator $[\phi^2]$ can be obtained by differentiating the partition function in the presence of $\mathcal{D}$ with respect to $g$ and striping off the integral over $\mathcal{D}$. Indeed, the operator insertion obtained in this way is finite and differs from the $\phi_0^2$ by an ascending series of poles in $\epsilon$. In principle, the contribution of the total derivatives to $[\phi^2]$ could be missed, because we explicitly strip off the integral over the defect. However, in our case total derivatives are not allowed by dimensional analysis. As a result, one gets

$$[\phi^2] = \frac{dg_0}{dg}\phi_0^2 + \frac{d\Lambda_0}{dg} - \frac{db_0}{dg}\frac{\mathcal{R}}{24\pi} \quad \Rightarrow \quad \phi_0^2 = \Big(\frac{dg_0}{dg}\Big)^{-1}\Big([\phi^2] - \frac{d\Lambda_0}{dg} + \frac{db_0}{dg}\frac{\mathcal{R}}{24\pi}\Big)\,. \qquad (52)$$

Combining, (46), (50), (51) and (52), yields

$$\hat{T} = \beta(g)[\phi^2] + \mathcal{A}\,, \qquad (53)$$

where $\mathcal{A}$ represents anomaly (identity operator), which is not essential for our needs. As expected, $\hat{T}$ is a finite operator, which needs no renormalization, and up to an anomaly term it vanishes at the UV and IR fixed points, $g_{UV} = 0$ and $g_{IR} = \frac{\epsilon}{\pi C}$.

Next, we use the general formula (31) to evaluate the difference between the anomaly coefficients at the UV and IR ends of the DRG flow on a spherical defect. In our example, $g_{UV} = 0$, thus the defect becomes trivial in the UV, and the anomaly vanishes. Substituting (46) into (31), yields[12]

$$
\begin{aligned}
b_{IR} &= -3\pi \int_{S^2} s^2(\sigma_1) \left\langle \hat{T}(\sigma_1)\hat{T}(0) \right\rangle \Big|_{R\to\infty} \\
&= -3\pi(d-4)^2 g_0^2 \left( \int_{S^2} s^2(\sigma_1) \left\langle \phi_0^2(\sigma_1)\phi_0^2(0) \right\rangle_0 - g_0 \int_{S^2}\int_{S^2} s^2(\sigma_1)\left\langle \phi_0^2(\sigma_1)\phi_0^2(\sigma_2)\phi_0^2(0)\right\rangle_0 \right)_{R\to\infty} \\
&\quad + \mathcal{O}(g_{IR}^4),
\end{aligned}
$$

(54)

where

$$
\begin{aligned}
\langle \phi_0(\sigma_1)\phi_0(\sigma_2) \rangle &= \frac{C_\phi}{s(\sigma_1,\sigma_2)^{d-2}}, \quad C_\phi = \frac{\Gamma\left(\frac{d-2}{2}\right)}{4\pi^{\frac{d}{2}}}, \\
\langle \phi_0^2(\sigma_1)\phi_0^2(\sigma_2) \rangle &= \frac{2C_\phi^2}{s(\sigma_1,\sigma_2)^{2(d-2)}}, \\
\langle \phi_0^2(\sigma_1)\phi_0^2(\sigma_2)\phi_0^2(\sigma_3) \rangle &= \frac{8C_\phi^3}{(s(\sigma_1,\sigma_2)s(\sigma_1,\sigma_3)s(\sigma_2,\sigma_3))^{d-2}}.
\end{aligned}
$$

(55)

In particular,

$$
C = 4C_\phi = \frac{1}{\pi^2}, \quad g_{IR} = \epsilon\pi.
$$

(56)

The two integrals within parenthesis in (54) can be evaluated in a closed form, see Appendix B. Substituting (B.5), (B.15) and (50), yields[13]

$$
b_{IR} = -\frac{3}{8\pi^3}\epsilon^2 g_{IR}^2 \left( \frac{\pi}{\epsilon} - \frac{2g_{IR}}{3\epsilon^2} \right) + \mathcal{O}(g_{IR}^4) = -\frac{\epsilon^3}{8} + \mathcal{O}(\epsilon^4).
$$

(57)

To check this result we perform an independent calculation of $b_{IR}$ based on the direct calculation of the $\mathcal{F}$-function. We have,

$$
-\mathcal{F} = \frac{g_0^2}{2}\int_{\mathcal{D}}\int_{\mathcal{D}} \left\langle \phi_0^2(\sigma_1)\phi_0^2(\sigma_2) \right\rangle_0 - \frac{g_0^3}{6}\int_{\mathcal{D}}\int_{\mathcal{D}}\int_{\mathcal{D}} \left\langle \phi_0^2(\sigma_1)\phi_0^2(\sigma_2)\phi_0^2(\sigma_3) \right\rangle_0 + \mathcal{O}(g_0^4).
$$

(58)

Substituting (50), (55) and using (B.5), (B.10) of Appendix B, we obtain

$$
-\mathcal{F} = \frac{g^2(\mu R)^{2\epsilon}}{2}\left(1 + \frac{2g}{\pi\epsilon}\right)\left(-\frac{1}{8\pi^2} + \mathcal{O}(\epsilon)\right) - \frac{g^3(\mu R)^{3\epsilon}}{6}\left(-\frac{2}{3\pi^3\epsilon} + \mathcal{O}(\epsilon^0)\right) + \mathcal{O}(g^4).
$$

(59)

This expression is not finite in the limit $\epsilon \to 0$, because we did not include the contribution of the geometric counterterm proportional to the integral of the Ricci scalar over the defect.[14]

---

[12]Note that for any $g$ along the RG trajectory, the expression for $S$ is manifestly finite in the limit $\epsilon \to 0$, therefore $\mathcal{O}(g^4)$ terms are free of poles in $\epsilon$, i.e., these corrections are at least $\mathcal{O}(\epsilon^4)$.

[13]This result agrees with (40) if the difference between the normalizations of $\phi_0^2$ and $\mathcal{O}_i$ is taken into account. It follows from (35) and (55) that one should use $C_1 = \sqrt{8}$ in (40) to compare with (57).

[14]It satisfies $b_0 = \frac{g^3}{24\pi^3\epsilon} + \mathcal{O}(g^4)$, because from (42) and (59), we have ($\Lambda_0 = 0$ in dimensional regularization),

$$
-\mathcal{F} = -\frac{g^3}{72\pi^3\epsilon} + \frac{b_0}{24\pi}\int_{\mathcal{D}} d^2\sigma \sqrt{\hat{\gamma}}\, \mathcal{R} + \mathcal{O}(g^4,\epsilon^0).
$$

This counterterm is a constant independent of $R$, and therefore it does not contribute to the RDE (25). Substituting (59) into (25), setting $\mu = R^{-1}$, and taking the limit $R \to \infty$, gives

$$b_{\text{IR}} = -\frac{\epsilon^3}{8} + \mathcal{O}(\epsilon^4),\tag{60}$$

in full agreement with (57).

## 5 Conclusions

In this paper we defined the *renormalised defect entropy* (RDE) (25) to characterise RG flows on the two-dimensional spherical defects embedded in a $d$-dimensional flat Euclidean bulk CFT. By definition, the RDE is finite along the entire RG flow from the UV to the IR fixed points. This construction is used to provide an alternative derivation of the sum rule (31), also known as the defect $b$-theorem [55]. More interestingly, we argue that the RDE is monotonically decreasing along the RG flows induced by weakly relevant deformations of the UV fixed point. This result is quite surprising considering the fact that monotonicity of the non-perturbative definition of RDE is not obvious.

The salient feature of the key identity (21) in our construction is that it can be generalized to higher dimensional spherical defects ($p > 2$). However, for such defects, the integral of the two-point function on the right hand side of (21) exhibits UV divergences. These divergences are closely related to the new type of non-universal terms appearing in the partition function for the higher dimensional defects. In particular, one has to modify the definition of RDE to isolate the subleading universal contribution associated with anomaly. The corresponding modification necessarily involves higher order derivatives of the partition function with respect to $R$, such that the non-universal terms are suitably removed. The final pattern for the higher dimensional RDE resembles the so-called renormalized entanglement entropy [63]. It includes the uncharted higher point correlators of the defect stress tensor, which make non-perturbative studies difficult. Even though it is hard to prove monotonicity or positivity of such constructions in general, it would be interesting to explore them further.

## Acknowledgments

We thank Zohar Komargodski and Avia Raviv-Moshe for helpful discussions and correspondence.

**Funding information** We are grateful to the Israeli Science Foundation Center of Excellence (grant No. 2289/18) for continuous support of our research. RS would like to thank SINP, Kolkata for hospitality during the completion of this work.

---

Hence, using (46) and (52), we recover the anomaly term in (53)

$$\mathcal{A} = (d-4)g_0 \Big(\frac{dg_0}{dg}\Big)^{-1} \frac{db_0}{dg} \frac{\mathcal{R}}{24\pi} = -\frac{\epsilon^3}{8} \frac{\mathcal{R}}{24\pi} + \mathcal{O}(\epsilon^4).$$

This result agrees with (57).

# A  Conformal Ward Identities in the presence of defects

For the sake of completeness of the presentation, in this Appendix we reproduce a slight generalization of the Ward identities obtained in [64] to account for defects of codimension higher than one. We apply the identities to the special case of a spherical defect and recover (5).

Following [64], we extend (2) to include an extra term coupled to the normal derivatives of the metric,

$$
\delta W = -\frac{1}{2}\int_{\mathcal{M}} d^d x \sqrt{g}\; \delta g_{\mu\nu}\langle T^{\mu\nu}\rangle + \int_{\mathcal{D}} d^p\sigma\sqrt{\hat{\gamma}}\;\delta X^\mu(\sigma)\langle D_\mu\rangle
$$
$$
-\frac{1}{2}\int_{\mathcal{D}} d^p\sigma\sqrt{\hat{\gamma}}\left[\delta g_{\mu\nu}\langle\hat{T}^{\mu\nu}\rangle + \nabla_I\delta g_{\mu\nu}\langle\hat{A}^{I\mu\nu}\rangle + ...\right]. \tag{A.1}
$$

In principle, there could be additional terms coupled to higher order normal derivatives of the metric. However, they vanish at the fixed points of DRG provided that $p < 4$. Moreover, even if the first additional term of this kind is present, we argue that the Ward identity (5) is not modified in the special case of spherical defect.

The bulk diffeomorphism $x^\mu \to x'^\mu = x^\mu - \xi^\mu$ yields

$$
\delta g_{\mu\nu} = \nabla_\mu\xi_\nu + \nabla_\nu\xi_\mu, \qquad \delta X^u = -\xi^\mu. \tag{A.2}
$$

Together with (A.1) it results in the following Ward identity,

$$
0 = \int_{\mathcal{M}} d^d x \sqrt{g}\;\xi_\nu\nabla_\mu T^{\mu\nu} - \int_{\mathcal{D}} d^p\sigma\sqrt{\hat{\gamma}}\left[\xi^\mu D_\mu + \nabla_\mu\xi_\nu\hat{T}^{\mu\nu} + \nabla_I\nabla_\mu\xi_\nu\hat{A}^{I\mu\nu}\right]. \tag{A.3}
$$

Thereafter, we split every vector $\xi_\nu$ into components that are tangential and normal to the defect, using the tangent frame $e_a^\mu = \frac{\partial X^\mu}{\partial\sigma^a}$ and a normal frame $n_I^\mu$.

$$
\xi_\nu = e_\nu^a\xi_a + n_\nu^I\xi_I, \qquad \nabla_\mu = e_\mu^a\nabla_a + n_\mu^I\nabla_I. \tag{A.4}
$$

In all further computations, we will make the following assumptions for the tangent and normal frames: $\nabla_I n_J = \nabla_I e_a = 0$. This assumption corresponds to a particular choice of foliation in the vicinity of the defect, because $n_J^\mu$ and $e_a^\mu$ become bulk fields. In the case of a generic two-dimensional defect embedded within a curved ambient space, this type of foliation might not necessarily exist. However, it can be readily established for a two-dimensional spherical defect in flat space, which is a central focus of our paper. To achieve this, one simply needs to adopt spherical coordinates in the three-dimensional ambient space that encloses the sphere, while employing Cartesian coordinates for the remaining $d - 3$ dimensions.[15] The Ward identities remain unaffected by the choice of foliation; hence, our results are applicable for all geometries that admit a foliation of the above type.

To evaluate the above expression, we make use of the following identities,

$$
\nabla_a e_\nu^b = -n_\nu^I K_{Ia}^b, \qquad \nabla_a n_\mu^I = K_{ab}^I e_\mu^b, \tag{A.5}
$$

---

[15]The flat metric in these coordinates takes the form $ds^2 = dr^2 + r^2(d\theta^2 + \sin^2\theta d\phi^2) + \delta^{ij}dx_i dx_j$. The defect is characterized by $r = R$ and $x^i = 0$. As a result, the normal frame extends into the bulk field with $n_r^\mu\partial_\mu = \partial_r$ and $n_i^\mu\partial_\mu = \partial_i$ for $i = 4, 5, ..., d$. Likewise, the tangent frame extends into the bulk field, giving $e_\phi^\mu\partial_\mu = \frac{R}{r}\partial_\phi$ and $e_\theta^\mu\partial_\mu = \frac{R}{r}\partial_\theta$. Upon direct calculation, we find $\nabla_r e_\theta^\mu = \nabla_r e_\phi^\mu = \nabla_r n_r^\mu = \nabla_r n_i^\mu = 0$, and similarly for the derivative in the direction of $n_i^\mu$. The useful non-zero Christoffel symbols are $\Gamma_{\theta,r\theta} = -\Gamma_{r,\theta\theta} = r$, $\Gamma_{\phi,r\phi} = -\Gamma_{r,\phi\phi} = r\sin^2\theta$, and $\Gamma_{\phi,\theta\phi} = -\Gamma_{\theta,\phi\phi} = \frac{1}{2}r^2\sin(2\theta)$.

where $K_{ab}$ are the extrinsic curvatures of the defect manifold $\mathcal{D}$. Using these identities, we obtain

$$\nabla_\mu \xi_\nu = e_\mu^a e_\nu^b \nabla_a \xi_b - e_\mu^a n_\nu^I K_{Ia}^b \xi_b + e_\mu^a n_\nu^I \nabla_a \xi_I + e_\mu^a e_\nu^b K_{ab}^I \xi_I + n_\mu^I e_\nu^b \nabla_I \xi_b + n_\mu^I n_\nu^J \nabla_I \xi_J \,, \tag{A.6}$$

$$\begin{aligned}
\nabla_I \nabla_\mu \xi_\nu &= e_\mu^a e_\nu^b \nabla_I \nabla_a \xi_b - e_\mu^a n_\nu^J \left( \xi_b \nabla_I K_{Ja}^b + K_{Ja}^b \nabla_I \xi_b \right) + e_\mu^a n_\nu^J \nabla_I \nabla_a \xi_J \\
&\quad + e_\mu^a e_\mu^b \left( \nabla_I K_{ab}^J \xi_J + K_{ab}^J \nabla_I \xi_J \right) + n_\mu^J e_\nu^b \nabla_I \nabla_J \xi_b + n_\mu^J n_\nu^K \nabla_I \nabla_J \xi_K \,. \tag{A.7}
\end{aligned}$$

Substituting equations (A.6),(A.7) into (A.3) yields,

$$\begin{aligned}
\delta W &= \int_{\mathcal{M}} d^d x \sqrt{g}\, \xi_\nu \nabla_\mu T^{\mu\nu} + \int_{\mathcal{D}} d^p \sigma \sqrt{\hat{\gamma}} \Big[ -D^a \xi_a - D^I \xi_I - \hat{T}^{ab} \nabla_a \xi_b + \hat{T}^{Ia} K_{Ia}^b \xi_b \\
&\quad - \hat{T}^{Ia} \nabla_a \xi_I - \hat{T}^{ab} K_{ab}^I \xi_I - \hat{T}^{Ia} \nabla_I \xi_a - \nabla_I \xi_J \hat{T}^{IJ} - \hat{A}^{Iab} \nabla_I \nabla_a \xi_b + \hat{A}^{IaJ} K_{Ja}^b \nabla_I \xi_b \\
&\quad - \hat{A}^{IaJ} \nabla_I \nabla_a \xi_J - \hat{A}^{Iab} K_{ab}^J \nabla_I \xi_J - \hat{A}^{IJb} \nabla_I \nabla_J \xi_b - \hat{A}^{IJK} \nabla_I \nabla_J \xi_K + \dots \Big] \,. \tag{A.8}
\end{aligned}$$

To proceed, change the order of $\nabla_I \nabla_a$ to $\nabla_a \nabla_I$ and then integrate by parts over the defect submanifold. Using the identity $\left[ \nabla_\mu, \nabla_\nu \right] \xi_\rho = -\mathcal{R}^\sigma_{\ \rho\mu\nu} \xi_\sigma = \mathcal{R}^{\ \ \sigma}_{\rho\ \mu\nu} \xi_\sigma$, one arrives at,

$$\begin{aligned}
\delta W &= \int_{\mathcal{M}} d^d x \sqrt{g}\, \xi_\nu \nabla_\mu T^{\mu\nu} \tag{A.9} \\
&\quad + \int_{\mathcal{D}} d^p \sigma \sqrt{\hat{\gamma}} \Big[ \left( \nabla_a \hat{T}^{ab} - D^b + \nabla_a + \hat{T}^{Ia} K_{Ia}^b + \hat{A}^{Iac} \mathcal{R}^{\ \ b}_{c\ aI} - \nabla_c \left( \hat{A}^{Iab} K_{Ia}^c \right) \right) \xi_b \\
&\quad + \left( \nabla_a \hat{T}^{Ia} - D^I - \hat{T}^{ab} K_{ab}^I + \hat{A}^{Jab} \mathcal{R}^{\ \ I}_{b\ aJ} + \hat{A}^{Jab} K_{Ja}^c K_{cb}^I \right) \xi_I \\
&\quad + \left( -\hat{T}^{Ia} + \nabla_b \hat{A}^{Iba} \right) \nabla_I \xi_a + \left( -\hat{T}^{IJ} - \hat{A}^{Iab} K_{ab}^J \right) \nabla_I \xi_J \\
&\quad - \hat{A}^{IJb} \nabla_I \nabla_J \xi_b - \hat{A}^{IJK} \nabla_I \nabla_J \xi_K + \dots \Big] \,.
\end{aligned}$$

Since $\xi_b, \xi_I$ and their normal derivatives are completely independent, their coefficients must separately vanish. This leaves us with the following Ward Identities,

$$\begin{aligned}
\nabla_b \hat{T}^{ba} - D^a + K_{Ib}^a \hat{T}^{Ib} - \nabla_c \left( K_{Ib}^c \hat{A}^{Iba} \right) + \mathcal{R}^{\ \ a}_{c\ bI} \hat{A}^{Ibc} &= 0 \,, \\
\nabla_a \hat{T}^{aI} - D^I - K_{ab}^I \hat{T}^{ab} + \left( \mathcal{R}^{\ \ I}_{b\ aJ} + K_{Ja}^c K_{cb}^I \right) \hat{A}^{Jab} &= 0 \,, \\
\nabla_a \hat{A}^{Iab} - \hat{T}^{Ib} &= 0 \,, \tag{A.10} \\
\hat{T}^{IJ} + \hat{A}^{Iab} K_{ab}^J &= 0 \,, \\
\hat{A}^{IJb} = \hat{A}^{IJK} &= 0 \,.
\end{aligned}$$

For a $p$-dimensional spherical defect and flat bulk with $\mathcal{R} = 0$, there is only one normal direction to the sphere with a non-vanishing extrinsic curvature, $K_{ab} = \frac{g_{ab}}{R}$. However, its contribution to the first equation in (A.10) vanishes with the use of the third equation. Thus we simply have

$$\nabla_b \hat{T}^{ba} = D^a \,. \tag{A.11}$$

# B Useful integrals

In this Appendix we evaluate various integrals on a $p$-dimensional spherical defect of radius $R$. These integrals are used in the main body of the text. It is convenient to describe the metric

on $S^p$ through the use of stereographic projection on $\mathbb{R}^p$. In these coordinates the metric is conformally flat,

$$\hat{\gamma}_{ac}dx^a dx^c = \frac{4R^2}{(1+|x|^2)^2}\delta_{ac}dx^a dx^c, \quad |x|^2 = \delta_{ac}x^a x^c. \tag{B.1}$$

In particular, the chordal distance between the two points on $S^p$ takes the form

$$s(x_1, x_2) = 2R\frac{|x_1 - x_2|}{(1+|x_1|^2)^{1/2}(1+|x_2|^2)^{1/2}}. \tag{B.2}$$

We start from the following double integral

$$I_1 = \int \prod_{i=1}^{2} d^p x_i \sqrt{\hat{\gamma}(x_i)}\frac{1}{[s(x_1, x_2)]^{2\alpha}}. \tag{B.3}$$

Note that the integral over $x_1$ is independent of $x_2$, because the integrand is invariant under rigid rotations of the sphere. Hence, we can set $x_2 = 0$ without changing the answer. As a result, we obtain

$$I_1 = \int \prod_{i=1}^{2} d^p x_i \sqrt{\hat{\gamma}(x_i)}\frac{1}{[s(x_1, 0)]^{2\alpha}} = R^{2(p-\alpha)}\frac{2^{1+p-2\alpha}\pi^{\frac{p+1}{2}}}{\Gamma\left(\frac{p+1}{2}\right)}\int \frac{d^p x_1}{(1+x_1^2)^{p-\alpha}|x_1|^{2\alpha}}, \tag{B.4}$$

where in the second equality we used (B.1), (B.2) and integrated over $x_2$. Using spherical coordinates to perform the remaining integral, yields

$$I_1 = R^{2(p-\alpha)}\frac{2^{1+p-2\alpha}\pi^{p+\frac{1}{2}}\Gamma\left(\frac{p}{2}-\alpha\right)}{\Gamma\left(\frac{p+1}{2}\right)\Gamma(p-\alpha)}. \tag{B.5}$$

Next we calculate

$$I_2 = \int \prod_{i=1}^{3} d^p x_i \sqrt{\hat{\gamma}(x_i)}\frac{1}{[s(x_1, x_2)s(x_2, x_3)s(x_3, x_1)]^{p-\epsilon}}. \tag{B.6}$$

Let us carry out the integrals over $x_1$ and $x_2$ first. Due to the manifest invariance of the integrand under the rotations of the sphere, the final result is independent of $x_3$, and therefore we can set $x_3 = 0$. Thus the integral over $x_3$ gives the volume of $S^p$,

$$I_2 = \frac{2\pi^{\frac{p+1}{2}}R^p}{\Gamma\left(\frac{p+1}{2}\right)}\int \prod_{i=1}^{2} d^p x_i \sqrt{\hat{\gamma}(x_i)}\frac{1}{[s(x_1, x_2)s(x_2, 0)s(0, x_1)]^{p-\epsilon}} \tag{B.7}$$

$$= (2R)^{3\epsilon}\frac{2^{1-p}\pi^{\frac{p+1}{2}}}{\Gamma\left(\frac{p+1}{2}\right)}\int \prod_{i=1}^{2} d^p x_i \frac{1}{(|x_{12}||x_2||x_1|)^{p-\epsilon}}\frac{1}{\left[\left(1+x_1^2\right)\left(1+x_2^2\right)\right]^{\epsilon}}, \quad x_{12} = x_1 - x_2.$$

To simplify the double integral, we apply inversion $|x_{1,2}| \to |x_{1,2}|^{-1}$,

$$I_2 = (2R)^{3\epsilon}\frac{2^{1-p}\pi^{\frac{p+1}{2}}}{\Gamma\left(\frac{p+1}{2}\right)}\int \prod_{i=1}^{2} d^p x_i \frac{1}{|x_{12}|^{p-\epsilon}}\frac{1}{\left[\left(1+x_1^2\right)\left(1+x_2^2\right)\right]^{\epsilon}}. \tag{B.8}$$

Using the standard Feynman parametrization to integrate over $x_1$, yields

$$I_2 = (2R)^{3\epsilon}\frac{2^{1-p}\pi^{\frac{2p+1}{2}}\Gamma\left(\frac{\epsilon}{2}\right)}{\Gamma\left(\frac{p+1}{2}\right)\Gamma\left(\frac{p-\epsilon}{2}\right)\Gamma(\epsilon)}\int d^p x_2 \int_0^1 du \frac{(1-u)^{\frac{p-\epsilon-2}{2}}u^{\frac{\epsilon}{2}-1}}{\left(1+x_2^2\right)^{\epsilon}\left(1+(1-u)x_2^2\right)^{\frac{\epsilon}{2}}}. \tag{B.9}$$

Integrating over the Feynman parameter $u$, and then using the spherical coordinates to carry out the remaining integral over $x_2$, we obtain

$$I_2 = R^{3\epsilon} \frac{8\pi^{\frac{3(p+1)}{2}} \Gamma\left(-\frac{p}{2} + \frac{3\epsilon}{2}\right)}{\Gamma(p)\Gamma\left(\frac{1+\epsilon}{2}\right)^3} . \tag{B.10}$$

Finally, we evaluate a triple integral of the form

$$I_3 = \int \prod_{i=1}^{3} d^p x_i \sqrt{\hat{\gamma}(x_i)} \frac{s(x_1, x_2)^2}{[s(x_1, x_2)s(x_2, x_3)s(x_3, x_1)]^{(p-\epsilon)}} . \tag{B.11}$$

As before, the rotational symmetry can be used to set $x_3 = 0$,

$$\begin{aligned}
I_3 &= \int \prod_{i=1}^{3} d^p x_i \sqrt{\hat{\gamma}(x_i)} \frac{s(x_1, x_2)^2}{[s(x_1, x_2)s(x_2, 0)s(0, x_1)]^{(p-\epsilon)}} \\
&= \frac{2^{1-p}\pi^{\frac{p+1}{2}}}{\Gamma\left(\frac{p+1}{2}\right)} (2R)^{3\epsilon+2} \int \prod_{i=1}^{2} d^p x_i \frac{1}{(1+x_1^2)^{1+\epsilon}(1+x_2^2)^{1+\epsilon} |x_{12}|^{p-\epsilon-2}|x_2|^{p-\epsilon}|x_1|^{p-\epsilon}} ,
\end{aligned} \tag{B.12}$$

where in the second equality we used (B.1), (B.2). Next, we apply inversion $|x_{1,2}| \to |x_{1,2}|^{-1}$ to simplify the double integral

$$I_3 = \frac{2^{1-p}\pi^{\frac{p+1}{2}}}{\Gamma\left(\frac{p+1}{2}\right)} (2R)^{3\epsilon+2} \int \prod_{i=1}^{2} d^p x_i \frac{1}{(1+x_1^2)^{1+\epsilon}(1+x_2^2)^{1+\epsilon} |x_{12}|^{p-\epsilon-2}} . \tag{B.13}$$

Introducing Feynman parametrization to integrate over $x_1$, yields

$$I_3 = (2R)^{3\epsilon+2} \frac{2^{1-p}\pi^{\frac{2p+1}{2}}\Gamma\left(\frac{\epsilon}{2}\right)}{\Gamma\left(\frac{p+1}{2}\right)\Gamma\left(\frac{p-\epsilon-2}{2}\right)\Gamma(\epsilon+1)} \int d^p x_2 \int_0^1 du \frac{(1-u)^{\frac{p-\epsilon-4}{2}} u^{\frac{\epsilon}{2}}}{\left(1+x_2^2\right)^{1+\epsilon}\left(1+(1-u)x_2^2\right)^{\frac{\epsilon}{2}}} . \tag{B.14}$$

Integrating now over the Feynman parameter $u$, and then using spherical coordinates to calculate the integral over $x_2$, we obtain

$$I_3 = (2R)^{3\epsilon+2} \frac{\pi^{3p/2+1} 2^{1-2\epsilon}\Gamma\left(\frac{3\epsilon}{2}+1-\frac{p}{2}\right)\Gamma(\epsilon/2)}{\Gamma(1+\epsilon)\Gamma\left(\frac{1+\epsilon}{2}\right)^2\Gamma(p)} . \tag{B.15}$$

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
