# Peer review of "RG flows on two-dimensional spherical defects"

_SciPost Physics, doi:SciPost Phys. 15, 240 (2023)_

## Round 1 · Referee Report · Anonymous (Referee 1) · 2023-10-18

Report

The authors addressed my observations in a satisfactory way. I am happy to recommend the paper for publication.

---

## Round 1 · Author Response

Dear Editor,

We express our sincere gratitude to the referee for their meticulous review of our manuscript. In the enclosed version of the paper, we have diligently addressed all of the concerns, comments, and suggestions provided by the referee.
We hope that the referee will find our replies to be satisfactory, and recommend the paper for publication in the journal.

Yours Sincerely,

The authors

---

## Round 1 · List of Changes

The referee's comments are withing quotation marks, followed by our replies.

  1. "In the introduction, the authors cite [53] as "motivating" their definitions, and this risks underrepresenting the debt of the present paper to that work. The authors should emphasize that the method of proof closely follows the one presented in [53]." In the current version, we rephrased this paragraph as follows:

-In what follows, we introduce a renormalized defect entropy which is fixed by the characteristic size of the defect. Our construction is similar to the one previously employed in the context of entanglement entropy [63]. For a DCFT, it reduces to the dimensionless "central charge" that multiplies the Euler density in the defect's Weyl anomaly, whereas for a general quantum field theory, it interpolates between the central charges of the UV and IR fixed points as the radius of the spherical defect is varied from zero to infinity. Using the ideas introduced in [53], we show that the renormalized defect entropy necessarily decreases from its initial value along the DRG flow, thus providing an alternative proof for irreversibility of the DRG flows on two-dimensional defects [55]. Furthermore, we argue that when the DRG flow is induced by a sufficiently weak relevant deformation of the UV fixed point, the renormalized defect entropy exhibits monotonic decrease and plays the role of a C-function throughout all orders in perturbation theory.

  1. "There is a notational ambiguity in section 2, which might be confusing. In 2.2, the defect contributions are integrated in d(sigma), and so the background fields, including the metric, depend on sigma through the spacetime coordinate x. However, in 2.6, the variation under reparametrization is taken to vanish. A consistent treatment requires to take the dependence on sigma into account. This should affect a number of equation, including the unnamed equation after 2.7, (hopefully) without affecting any of the physical consequences."

-This variation is zero by definition (the variation = difference between the new and old fields at the same point). We added a footnote 3, which hopefully addresses this confusion.

3." After 3.16, there is a wrong reference to eq. 4.10 (it is more natural to refer to 3.15)."

-Fixed.

4."The symbol C is used with two different meanings in sections 3.1 and 4, which leads to a mismatch between eq. 3.18 and 4.16. The authors explain the mismatch in a footnote, but it would be better to use consistent definitions for C and either change the coefficient in 4.8 to C x (explicit normalization factor), or use a different symbol, e.g. \tilde{C}."

-Fixed, see (3.19).

5."After 3.19, the authors claim that the entropy function is monotonic at all orders in the coupling, but they only proved the statement for the leading order. There is a difference between the two statements: what if the coupling is only marginally relevant? The authors should prove the statement more generally in the case where the beta function starts at higher orders, or they must limit their claim appropriately."

-Indeed, we compute the derivative of S to the leading order along the RG flow. In the case of weakly relevant deformations, this requires accounting for both quadratic and cubic terms in the coupling constants (the quadratic term is accompanied by an additional factor of epsilon, rendering it comparable to the cubic correction as we approach the IR fixed point). Our computation demonstrates a negative sign for the derivative, implying the monotonic nature of S. Higher-order corrections cannot alter this conclusion (change the sign of derivative), as they remain smaller than the leading-order outcome, otherwise perturbation theory would lack any sense.

Alternatively, the higher order corrections change h_ij in (3.20), but it remains perturbatively close to the identity matrix, and therefore h_ij retains its positive definiteness. We rephrased our wording to emphasize that our analysis does not include the case of marginally relevant deformations.

6." In appendix A, the description of the defect in terms of a system of coordinates sigma is chosen. Then, the tangent and normal vectors and the curvatures of the defect are functions of these coordinates and only defined at the defect. Therefore, their derivatives in directions orthogonal to the defect are not defined, and cannot be taken to vanish as after A.4. The authors can for instance compare with refs [23] and [29], where these higher order terms are taken into account, to rephrase their discussion when necessary. They can also switch to a definition of defect geometry in terms of an adapted foliation of the full spacetime, in which case the normal derivatives mentioned above make sense and must be included in the computation."

-We have tacitly assumed the existence of a foliation adhering to the specified characteristics. In simpler terms, we have extended the normal and tangent bundles to the bulk fields, thereby making the derivatives in directions orthogonal to the defect well-defined. A foliation of this type exists for any co-dimension 1 defect or surface. However, for higher codimensions, the existence of such a foliation is not guaranteed. This is precisely why we have explicitly articulated this as our underlying assumption. It is important to note that this assumption holds for the geometry studied in our paper (a sphere in flat space), and therefore the formulas obtained are sufficient for our goals. In cases where constructing such a foliation is not possible, potential additional corrections may come into play. Nevertheless, setups of this kind fall beyond the scope of our paper, rendering it impractical to derive useless formulas. For enhanced clarity, we have incorporated a dedicated paragraph below section (A.4) and introduced footnotes 2 and 15, which furnishes a comprehensive explanation of our assumption.

---

## Editorial Decision

published